# Major Left Bundle Branch Block and Coronary Heart Disease—Are There Any Differences between the Sexes?

**DOI:** 10.3390/jcm10112284

**Published:** 2021-05-25

**Authors:** Diana Gurzău, Alexandra Dădârlat-Pop, Bogdan Caloian, Gabriel Cismaru, Horaţiu Comşa, Raluca Tomoaia, Dumitru Zdrenghea, Dana Pop

**Affiliations:** “Iuliu Hațieganu”, University of Medicine and Pharmacy, 400012 Cluj-Napoca, Romania; gurzaudiana@yahoo.com (D.G.); dadarlat.alexandra@yahoo.com (A.D.-P.); gabi_cismaru@yahoo.com (G.C.); dh.comsa@gmail.com (H.C.); raluca.tomoaia@gmail.com (R.T.); dzdrenghea@yahoo.com (D.Z.); pop67dana@gmail.com (D.P.)

**Keywords:** LBBB, ischemic heart disease, echocardiography, coronary angiography, gender differences

## Abstract

Left bundle branch block is not a benign pathology, and its presence requires the identification of a pathological substrate, such as ischemic heart disease. Left bundle branch block appears to be more commonly associated with normal coronary arteries, especially in women. The objectives of our study were to describe the particularities of left bundle branch block in women compared to men with ischemic heart disease. Result: We included seventy patients with left bundle branch block and ischemic heart disease, with a mean age of 67.01 ± 8.89 years. There were no differences in the profile of risk factors, except for smoking and uric acid. The ventricular depolarization (QRS) duration was longer in men than women (136.86 ± 8.32 vs. 132.57 ± 9.19 msec; *p* = 0.018) and also men were observed to have larger left ventricular diameters. Left bundle branch block duration was directly associated with ventricular diameters and indirectly associated with left ventricular ejection fraction value, especially in women (*R* = −0.52, *p* = 0.0012 vs. *R* = −0.50, *p* = 0.002). In angiography, 80% of women had normal epicardial arteries compared with 65.7% of men; all these patients presented with microvascular dysfunction. Conclusion: The differences between the sexes were not so obvious in terms of the presence of risk factors; instead, there were differences in electrocardiographic, echocardiographic, and angiographic aspects. Left bundle branch block appears to be a marker of microvascular angina and systolic dysfunction, especially in women.

## 1. Introduction

The presence of left bundle branch block (LBBB), although extensively studied, is still associated with many gaps in the knowledge of its development mechanism and underlying associated diseases. Major left bundle branch block is not a benign electro-physiological condition. On the contrary, it almost always has an underlying disease: hypertension, left ventricular hypertrophy, cardiomyopathies, valvulopathies, heart failure, or ischemic heart disease. However, LBBB may also occur in young patients, without any structural heart disease, being associated with a good prognosis [1]. The presence of bundle branch block in older patients with multiple associated comorbidities has proven to be an independent negative predictor of cardiovascular events [2].

However, most of the time, LBBB is associated with myocardial ischemia, through both obstructive coronary artery disease and microvascular involvement, or with coronary vasospasm. Electrocardiographic recording of this conduction disorder in a suggestive clinical context requires establishing the diagnosis of coronary disease or even acute myocardial infarction. Thus, new-onset LBBB associated with an increase in myocardial necrosis enzyme levels defines the diagnosis of acute myocardial infarction, according to the European Society of Cardiology [3].

The presence of LBBB is associated with a negative prognostic role, and in patients with ischemic heart disease it is correlated with a higher mortality rate [2]. In women, it seems that the presence of LBBB induces a higher incidence of heart failure as well as mortality from this disease [4,5]. In men, LBBB is a predictor of mortality from coronary disease [4,5]. In the presence of this conduction disorder, abnormal ventricular activation occurs, with the development in time of structural changes as well as regional myocardial perfusion abnormalities [6].

There are very few studies regarding the presence of LBBB in women with ischemic heart disease, particularly in the case of microvascular involvement. Thus, in this context, the objectives of the current study were to describe certain particularities of LBBB in women compared to men with ischemic heart disease, especially from the point of view of echocardiography and coronary angiography.

## 2. Materials and Methods

The study included seventy patients admitted to the Cardiology Department of the Clinical Rehabilitation Hospital, Cluj-Napoca, Romania, with major LBBB detected on the resting electrocardiogram recorded on the first day of admission and symptoms highly suggestive of myocardial ischemia. Fifty percent of these patients were women, with a mean age of 67.01 ± 8.89 years. All patients were assessed for demographic and clinical characteristics as well as electrocardiographic and ultrasound parameters. Regarding the ultrasound parameters, the ventricular diameters (which were measured from the parasternal long axis view), the presence of interventricular asynchrony, and diastolic and systolic function were evaluated. The Simpson’s biplane formula was used to calculate the left ventricular ejection fraction (LVEF%). We note that all patients included in the study underwent invasive coronary assessment via angiography for a high index of suspicion for ischemic coronary heart disease. Based on the coronary angiography results, we defined ischemic heart disease as macrovascular in the presence of atherosclerotic lesions of the epicardial arteries >70% or >50% in the left main/proximal left anterior descending artery, and microvascular angina when the epicardial arteries were angiographically normal but with a decreased epicardial flow of the contrast media [7,8]. In terms of comorbidities, we evaluated the presence of obesity, dyslipidemia, hypertension, diabetes, atrial fibrillation, and heart failure. The diagnosis of heart failure was established based on symptoms, the value of NT-proBNP (the value >125 pg/dL being considered diagnostic), and diastolic and/or systolic dysfunction, including both patients with reduced ejection fraction (<50%) as well as those with preserved left ventricular ejection fraction. LBBB was diagnosed in the presence of a wide QRS complex, with a duration ≥120 msec, with a deep and wide S wave in V1/V2, and a wide and tall R wave in leads V5/V6 [4]. Statistical analysis was performed in Writer, Presentation and Spreadsheets (WPS) Office 2019. For all statistical analyses and subsequent diagrams, R 4.0.0 was used. We considered that *p* < 0.05 was statistically significant and *p* < 0.10 only showed a tendency to statistical significance.

The selected patients were informed about the study protocol and signed an informed consent. The present clinical study was approved by the local ethics committee (approval number 2606/4 April 2018) and was performed in accordance with the ethical standards established by the 1964 Declaration of Helsinki and its later amendments.

## 3. Results

The general characteristics of the patients included in the study are synthesized in Table 1.

As can be seen in Table 1, there were no significant differences between the analyzed cardiovascular risk factors, except for smoking and uric acid value. Smoking was much more frequently found among men compared to women: 97.1% vs. 28.6%, respectively, *p* < 0.001. Also, a lower uric acid value was recorded in women (6.43 ± 1.68 vs. 7.73 ± 2.09, respectively, *p* = 0.021).

In terms of symptoms for ischemic heart disease, although it is known that women have atypical angina more frequently, in our study, the results were somewhat surprising. Women presented with a higher percentage of typical symptoms, both in the whole group of women and also compared to male patients, 30 (85.7%) vs. 22 (62.9%), respectively, *p* = 0.054.

Troponin was dosed at a slightly higher rate in women, 18.57% versus 15.71% for men, but without significant statistically differences. Both women and men had normal troponin values in a higher percentage compared to its reacted or increased values. Of all patients, 21.4% had an indication for natriuretic peptide sampling, in the context of typical symptoms of heart failure, with the mean NT-proBNP value being 2514.07 ± 1871.2 pg/mL. There were no differences between men and women regarding the value of this specific marker of heart failure.

Also, as can be observed, there were no differences regarding the various types of LBBB. In contrast, in women, the duration of the QRS complex was statistically significantly shorter than in men: 132.57 ± 9.19 vs. 136.86 ± 8.32 msec, respectively, *p*-0.018.

From an echocardiographic point of view, women had significantly smaller sizes of ventricular diameters, both systolic (37.91 ± 10.5 vs. 45.14 ± 11.5 mm, respectively, *p* = 0.004) and diastolic (53.77 ± 8.0 vs. 59.20 ± 9.83 mm, respectively, *p* = 0.014).

Regarding the associated comorbidities, women had a higher tendency to be hypertensive, diabetic, and obese, while men more frequently presented atrial fibrillation and heart failure, but without the presence of statistically significant differences.

From an angiographic point of view, it was found that LBBB in women was more frequently associated with the presence of normal epicardial arteries compared to male patients: 28 (80.0%) vs. 23 (65.7%), respectively (*p* = 0.282). For this difference to be statistically significant, a larger group of patients would be needed. All the patients with normal epicardial coronary arteries presented with microvascular disfunction detected by angiography (Table 2, Figure 1).

No statistical correlation was detected between the width of the QRS complex in LBBB and the magnitude of coronary artery involvement (number of lesions and affected vessels) among patients with ischemic heart disease through macrovascular involvement (Table 3, Figure 2).

No correlation could be established between the QRS duration and the presence of ventricular asynchrony, due to the fact that 97.1% of the patients included in the study had associated interventricular septal asynchrony/dyssynchrony. It is known that the presence of LBBB, through the inter- and intraventricular asynchrony that it induces, leads in time to the deterioration of left ventricular structure and function. It is important, however, to establish whether this deterioration is associated with the QRS complex duration, and in this case also with the severity of ischemia. Thus, we found a statistically significant correlation between the QRS complex duration and the left ventricular systolic and diastolic diameters assessed by echocardiography. Table 4 and Figure 3 and Figure 4 illustrate the mentioned variables and correlations.

Figure 5 illustrates the correlation between the QRS complex duration and the left ventricular ejection fraction in the general population (*R* = −0.52, *p* < 0.0001), and Figure 6 presents the correlation between the QRS complex duration and the left ventricular ejection fraction depending on sex. This is inversely proportional to the QRS complex duration: the longer the QRS complex duration, the lower the ejection fraction.

In our analysis, patient prognosis was assessed based on the development of heart failure, age, sex, and associated comorbidities (hypertension, diabetes mellitus, obesity, or the presence of associated atrial fibrillation). None of these parameters had statistical significance (Table 5 and Figure 7).

## 4. Discussion

The implications of the presence of major LBBB are still uncertain, which is due to the highly variable results of the existing studies and research related to this topic. In the current study, by comparatively analyzing the profile of male and female patients with major LBBB and coronary disease, we observed no significant differences regarding the presence of cardiovascular risk factors, except for smoking, which was found in a statistically significantly higher proportion of men compared to women. The gender difference in smoking rates is known worldwide; the World Health Organization estimates that about 40% of men smoke compared to only 9% of women, but the percentage is increasing among women [9]. So far, there are no studies that correlate smoking with the presence of major LBBB; the mechanisms through which smoking might influence the development of this intraventricular conduction disease, however, would rather be secondary to the consequences of smoking (i.e., the presence of coronary atherosclerotic disease and endothelial dysfunction) [10].

It is well known that an increased uric acid value is associated with a high cardiovascular risk, which favors hypertension, atherosclerotic disease, or other cardiovascular diseases [11,12]. The mechanism through which uric acid increases the risk of cardiovascular diseases is mainly represented by endothelial dysfunction resulting from oxidative stress with the reduction of nitric oxide, cytokine release, and the activation of platelets as well as endothelial cells from the vascular system, which thus stimulate and enhance the atherosclerotic process [13]. Men usually have higher uric acid values compared to women—a finding also observed in our study. There are data demonstrating that a high uric acid value is correlated with a higher risk of cardiovascular events and with higher rates of mortality, particularly from ischemic coronary disease in women, although not in men [14]. Thus, an increased uric acid value in women would rather explain the presence of cardiovascular disease, probably with a higher risk for secondary major LBBB also increasing the risk of cardiac events.

In the current study, there were no statistically significant differences between the type of bundle branch block and patient’s sex, but when analyzing the QRS complex duration, in male patients, the duration of the complex was longer compared to the female patients. There have been few literature studies evaluating the electrocardiographic differences of LBBB between the two sexes, but the few studies that exist show that normally the duration of the QRS complex is longer in men due to the larger body surface area as well as to the larger size of the heart, which also explains the differences between the echocardiographic characteristics in the two sexes: men have larger systolic and diastolic diameters than women [15,16]. This particularity is not entirely due to the differences in the body surface and the size of the heart, because when adjustments were made for patients’ weight and height, this hypothesis was not validated. There are probably other unknown intrinsic factors that contribute to the higher values of the QRS complex duration in men [17].

The results of our study demonstrate an up to 10 msec ± 10% shorter duration of the QRS complex in women, which is in accordance with the results reported in the literature [15]. In fact, in women, LBBB, although narrower, is more real compared to men, being accompanied by greater interventricular septal dyssynchrony. This is why a better response to resynchronization therapy is described in women, despite a shorter duration of the QRS complex [17,18].

So far, there have been no studies assessing the differences in the characteristics of LBBB in women compared to men, in the presence of ischemic heart disease. Studies evaluating the response to resynchronization therapy have suggested that the presence of LBBB with a wider QRS complex in women can be a marker of more advanced cardiac involvement, with extensive fibrosis and scarring of the myocardium and implicitly of the interventricular septum [15,19,20]. This latter aspect is in accordance with the results of our study, because at the time when we analyzed the presence of a relationship between the width of the QRS complex and the magnitude of coronary involvement, no statistical correlation was found. This is explained on the one hand by the low proportion of patients with severe coronary lesions inducing extensive fibrotic and scarring changes, so, implicitly, a wide QRS complex, particularly among women. In women, the longest duration of the QRS complex was 160 msec (in one case), with a mean of only 130 msec. Due to the small number of patients included in the study, the result was not statistically significant, but it seems that LBBB in women is more frequently associated with the presence of normal epicardial arteries than with the presence of coronary atherosclerotic disease, compared to men.

A study evaluating the angiographic characteristics of symptomatic patients with major LBBB concluded that this is rather correlated with normal epicardial arteries; more precisely, 64% of patients had angiographic results without changes [21,22,23], which was also demonstrated in our research. However, this study, as well as most of the literature studies addressing this subject, does not describe the differences between the two sexes from the point of view of the grade or type of coronary involvement. Thus, our study is one of the few that demonstrates a link between LBBB in a clinical context highly suggestive of myocardial ischemia and the more frequent presence of angiographically normal epicardial arteries in women. Very importantly, all the patients who presented with normal angiographic epicardial arteries had coronary microcirculation dysfunction, and this microvascular angina is known to be more common in females. Therefore, the major LBBB requires the evaluation of microvascular dysfunction and subsequent specific treatment, microvascular angina not being a pathological entity as benign as previously thought, and the association of the LBBB, by its consequences, only aggravates the prognosis of these patients.

The literature describes the presence of LBBB secondary to coronary atherosclerotic disease more frequently among male patients, and variables such as advanced age, the presence of diabetes mellitus, and an ejection fraction lower than 50% have been described as predictors of coronary disease and as prognostic factors in these patients [24,25]. The results obtained in our study are not in accordance with those previously mentioned, and age, sex, as well as associated comorbidities do not seem to influence the presence of coronary atherosclerotic disease or patient prognosis. Indeed, this finding can be due to the small number of patients included in the study.

Also, the literature has postulated the idea that major LBBB that occurs at a heart rate (HR) < 125 beats/minute and has an angina clinical picture is an indicator of coronary disease. On the other hand, this hypothesis was rejected by Hertzeanu et al., who maintain that major LBBB that occurs at a HR < 125/min cannot be per se a criterion for coronary disease [26,27]. The results of our study support the idea of Hertzeanu et al., through the fact that major LBBB with a HR < 125/min was not associated with the presence of coronary atherosclerotic lesions. In addition, in the group of these patients with macrovascular lesions, there was no correlation between LBBB with a HR < 125/min and the severity of coronary lesions. Consequently, this result suggests that regardless of sex, the presence of LBBB dependent on heart rate does not necessarily involve the presence of underlying coronary atherosclerotic disease, despite angina symptoms. In our study, only one case of conduction disorder at a HR > 125/min was described, and in this case, angina symptoms can be explained on the one hand by the increase in lactate concentration in the coronary sinus, with an improvement following nitrate administration. On the other hand, it can be explained by the presence of a dyssynergic contraction that stimulates the mechanoreceptors situated in the chest, which causes an angina clinical picture [27,28].

Regardless of the type of coronary involvement, microvascular angina or coronary atherosclerotic disease, the association of LBBB through the previously mentioned mechanisms contributes to the progressive deterioration of ventricular function [29,30]. However, it is important to establish whether this deterioration of ventricular structure and function is correlated in the first place with the duration of the QRS complex and not least with the severity of ischemic lesions. Our study demonstrated a statistically significant moderate correlation between QRS complex duration and left ventricular systolic and diastolic diameters in both sexes. The left ventricular ejection fraction was moderately correlated with the duration of the QRS complex, having statistical significance. The longer the QRS complex duration of the major LBBB, the lower the ejection fraction. This moderate correlation has statistical significance in the general study population, as well as depending on patient’s sex. A low ejection fraction is associated with structural changes and also with a considerable amount of fibrosis in the left ventricle. Therefore, one of the consequences of these alterations is the prolongation of the QRS complex duration.

Regarding this correlation depending on sex, although it is statistically significant for both sexes, it seems that in women there is a stronger correlation between the width of the QRS complex in major LBBB and the reduction of the ejection fraction. This could be explained by the fact that in women, bundle branch block is more suggestive, with more correct criteria compared to men; consequently, interventricular asynchrony should also be greater, contributing to the reduction of systolic function. The width or the duration of the QRS complex in major LBBB could be a marker for the estimation of systolic ventricular dysfunction.

## 5. Conclusions

In patients with major LBBB and ischemic heart disease, there were no significant differences between the sexes in terms of the profile of cardiovascular risk factors, with the exception of smoking and uric acid values. Instead, differences between the two sexes were recorded in terms of electrocardiography, echocardiography, and angiography. Men have a longer QRS complex duration and larger ventricular diameters than women, but it appears that in women LBBB is a more accurate marker of ventricular systolic dysfunction and microvascular angina compared to men.

## Figures and Tables

**Figure 1 jcm-10-02284-f001:**
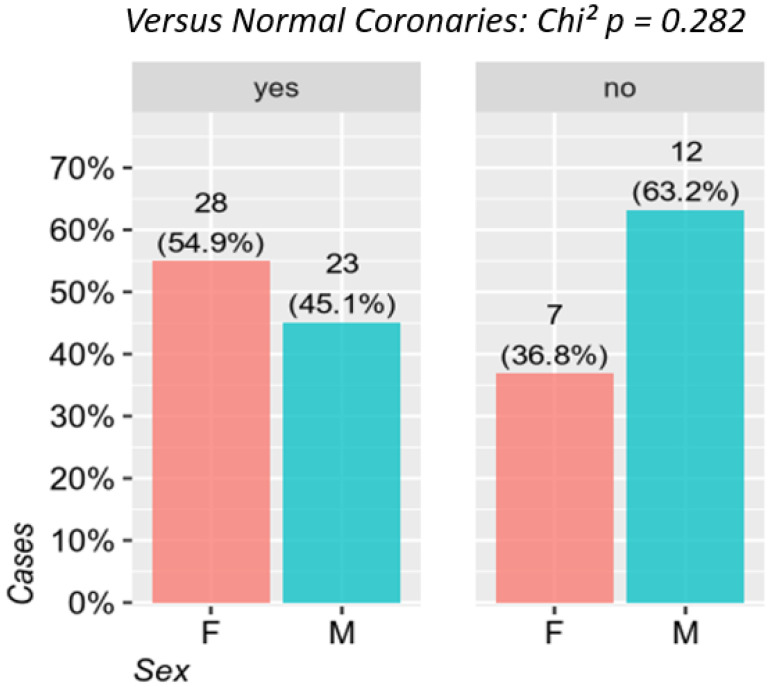
The relationship between the presence of LBBB and the normal coronary arteries between the two sexes. F—females, M—males.

**Figure 2 jcm-10-02284-f002:**
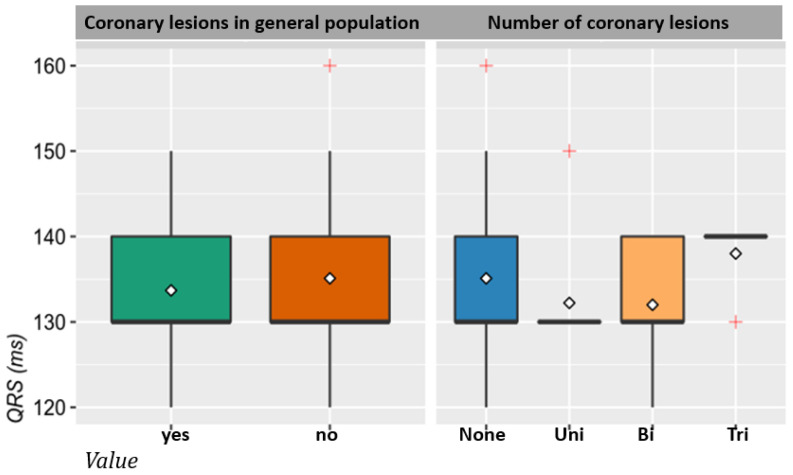
Correlation between the duration of left bundle branch block and macrovascular coronary lesions.

**Figure 3 jcm-10-02284-f003:**
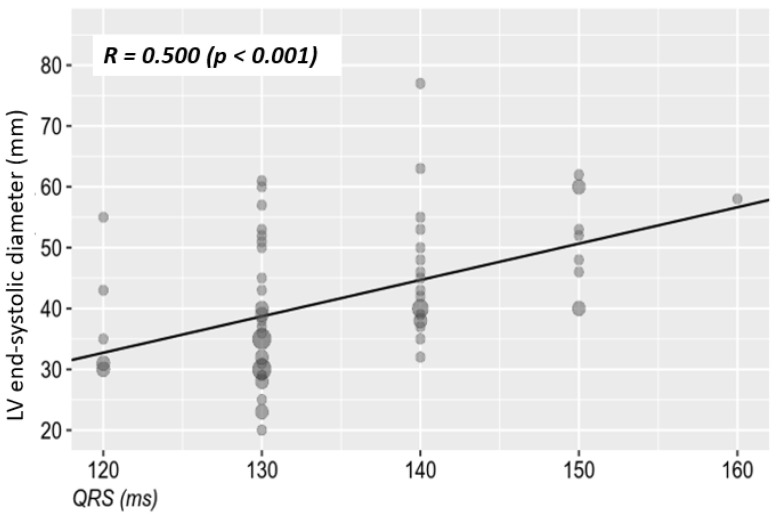
The relationship between the LBBB-QRS complex duration and systolic diameter of the left ventricle. LV—left ventricle.

**Figure 4 jcm-10-02284-f004:**
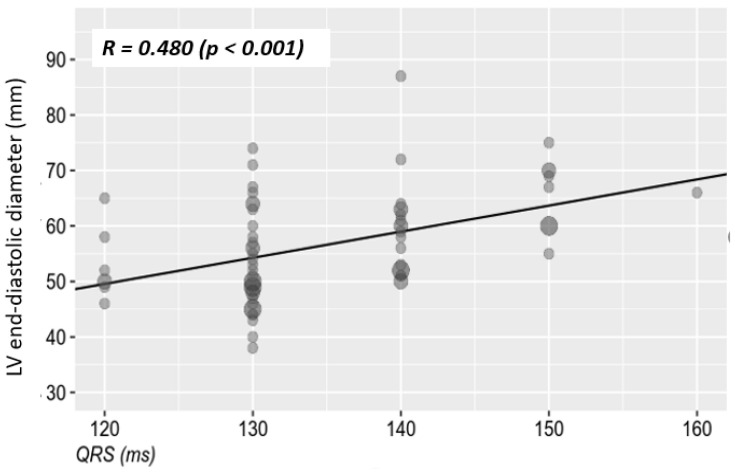
The relationship between the LBBB-QRS complex duration and the diastolic diameter of the left ventricle.

**Figure 5 jcm-10-02284-f005:**
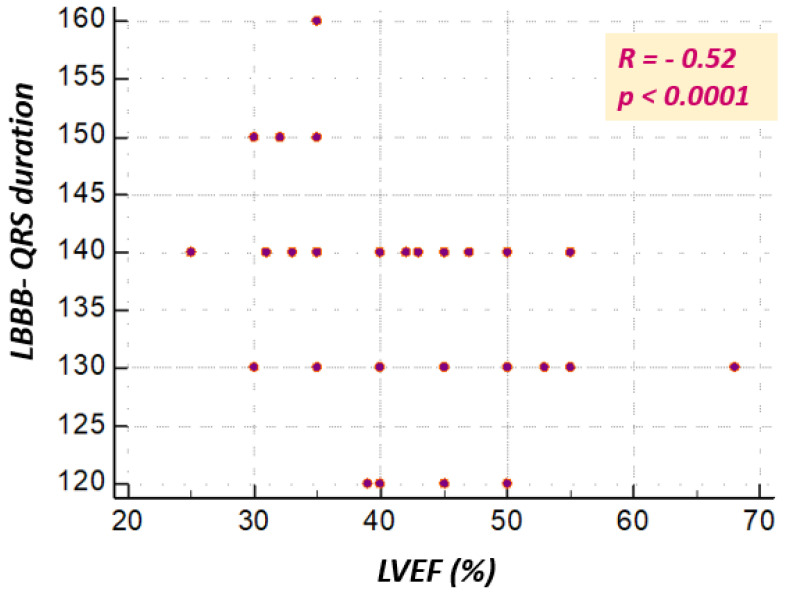
Correlation between LBBB-QRS complex duration and LVEF (%) in the general study population.

**Figure 6 jcm-10-02284-f006:**
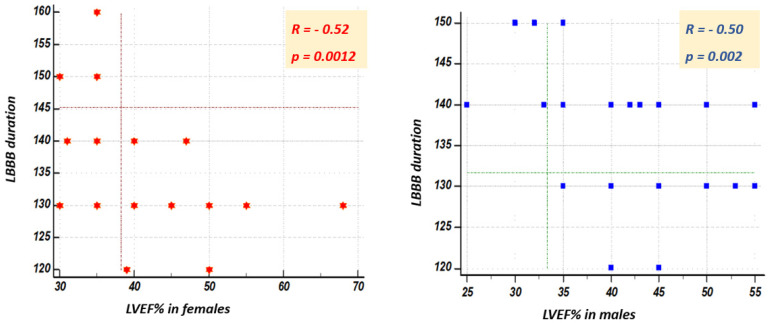
Correlation between the duration of the LBBB-QRS complex and LVEF (%) depending on the gender of the patient.

**Figure 7 jcm-10-02284-f007:**
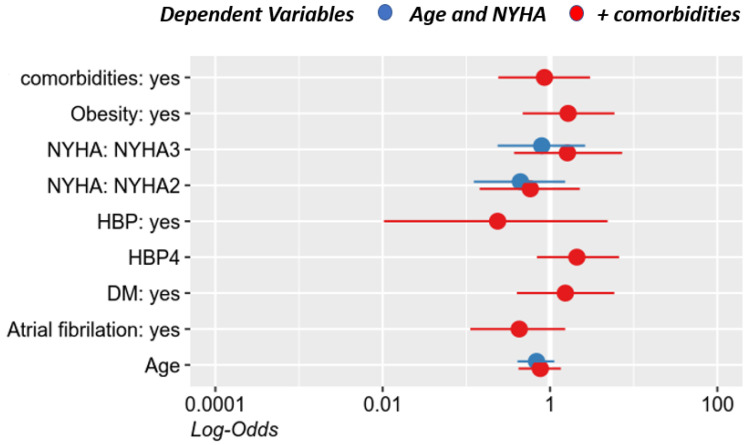
The prognosis of patients with ischemic heart disease and LBBB depending on the associated comorbidities. HBP: high blood pressure; HBP4: grades of hypertension; DM: diabetes mellitus.

**Table 1 jcm-10-02284-t001:** The main characteristics of the patients included in the study.

Variable	Details	Women	Men	Total	Statistics
Gender		35 (50.0%)	35 (50.0%)	70	
Age	*M* (min:max) μ ± *SD*	66 (47:88) 65.43 ± 8.51	69 (48:87) 68.60 ± 9.1	67 (47:88) 67.01 ± 8.89	*T*-test: *p* = 0.137
Smoking		10 (28.6%)	34 (97.1%)	44 (62.9%)	OR = 0.004 (0.00, 0.10, *p* < 0.001)
HR (b/min)	*M* (min:max) μ ± *SD*	70 (55:120) 72.80 ± 14.8	72 (50:145) 74.20 ± 16.8	71 (50:145) 73.50 ± 15.7	MW: *p* = 0.616
Systolic blood pressure (mmHg)	*M* (min:max) μ ± *SD*	130 (110:190) 132.14 ± 16.7	130 (100:180) 133.14 ± 20.0	130 (100:190) 132.64 ± 18.3	MW: *p* = 0.797
Diastolic blood pressure (mmHg)	*M* (min:max) μ ± *SD*	80 (60:100) 77.43 ± 12.2	80 (60:100) 79.29 ± 10.6	80 (60:100) 78.36 ± 11.4	MW: *p* = 0.363
Arterial hypertension		31 (88.6%)	30 (85.7%)	61 (87.1%)	OR = 1.29 [0.32, 5.28] (*p* > 0.999)
Diabetes mellitus		15 (42.9%)	10 (28.6%)	25 (35.7%)	OR = 1.87 [0.69, 5.06] (*p* = 0.318)
Obesity		28 (80.0%)	21 (60.0%)	49 (70.0%)	OR = 2.67 [0.92, 7.77] (*p* = 0.117)
HDL-c (mg/dL)	*M* (min:max) μ ± *SD*	42 (25:57) 40.94 ± 7.87	38 (25:69) 40.51 ± 10.1	40 (25:69) 40.73 ± 9.0	MW: *p* = 0.441
Triglycerides (mg/dL)	*M* (min:max) μ ± *SD*	161 (60:472) 168.37 ± 77.0	135 (47:473) 145.71 ± 69.3	150.5 (47:473) 157.04 ± 73.6	MW: *p* = 0.121
Glucose levels (mg/dL)	*M* (min:max) μ ± *SD*	109 (85:247) 121.51 ± 41.1	101 (82:225) 108.69 ± 28.4	102.5 (82:247) 115.10 ± 35.7	MW: *p* = 0.105
Uric acid	*M* (min:max) μ ± *SD*	6.32 (2.78:9.95) 6.43 ± 1.68	7.1 (4.64:11.45) 7.73 ± 2.09	6.65 (2.78:11.45) 7.08 ± 1.99	MW: *p* = 0.021
Typical angina		30 (85.7%)	22 (62.9%)	52 (74.3%)	OR = 3.55 [1.10, 11.41] (*p* = 0.054)
Atypical chest pain		4 (11.4%)	6 (17.1%)	10 (14.3%)	OR = 0.62 [0.16, 2.44] (*p* = 0.734)
Troponins	Elevated Normal value	1 (7.7%) 12 (92.3%)	2 (18.2%) 9 (81.8%)	3 (12.5%) 21 (87.5%)	OR = 0.38 [0.03, 4.81] *p* = 0.576
NT-proBNP > 125 pg/mL		6 (17.1%)	9 (25.7%)	15 (21.4%)	OR = 0.60 [0.19, 1.91] (*p* = 0.561)
NT-proBNP (pg/mL)	*M* (min:max) μ ± *SD*	2164 (300:7735) 2719.50 ± 2674.5	2071 (1107:4602) 2377.11 ± 1266.5	2071 (300:7735) 2514.07 ± 1871.2	MW: *p* = 0.955
QRS (msec)	*M* (min:max) μ ± *SD*	130 (120:160) 132.57 ± 9.19	140 (120:150) 136.86 ± 8.32	130 (120:160) 134.71 ± 8.96	MW: *p* = 0.018
LBBB	- new onset - permanent - intermittent - rate dependent	12 (34.3%) 22 (62.9%) 1 (2.9%) 0	8 (22.9%) 24 (68.6%) 2 (5.7%) 1 (2.9%)	20 (28.6%) 46 (65.7%) 3 (4.3%) 1 (1.4%)	V = 0.18 (*p* = 0.528)
LVEDD (mm)	*M* (min:max) μ ± *SD*	52 (38:70) 53.77 ± 8.0	58 (40:87) 59.20 ± 9.83	55.5 (38:87) 56.49 ± 9.31	*T*-test: *p* = 0.014
LVESD (mm)	*M* (min:max) μ ± *SD*	36 (23:60) 37.91 ± 10.5	43 (20:77) 45.14 ± 11.5	40 (20:77) 41.53 ± 11.5	MW: *p* = 0.004
LVEF %	*M* (min:max) μ ± *SD*	47 (30:68) 43.86 ± 8.9	43 (25:55) 41.94 ± 8.72	45 (25:68) 42.90 ± 8.8	MW: *p* = 0.488
Atrial fibrillation		8 (22.9%)	13 (37.1%)	21 (30.0%)	OR = 0.50 [0.18, 1.43] (*p* = 0.297)
Heart failure		23 (65.7%)	27 (77.1%)	50 (71.4%)	OR = 0.57 [0.20, 1.63] (*p* = 0.428)
NYHA classification	II III	9 (39.1%) 14 (60.9%)	14 (51.9%) 13 (48.1%)	23 (46.0%) 27 (54.0%)	OR = 0.60 [0.19, 1.84] (*p* = 0.407)

HR: heart rate; LBBB: left bundle branch block; LVEDD: left ventricular end-diastolic diameter; LVESD: left ventricular end-systolic diameter; LVEF %: left ventricular ejection fraction; NYHA: New York Heart Association.

**Table 2 jcm-10-02284-t002:** Association of LBBB with the presence of normal coronary arteries detected by angiography.

Variable	Details	Female	Male	Total	Statistics
Gender		35 (50%)	35 (50%)	70	
Normal coronary arteries		28 (80.0%)	23 (65.7%)	51 (72.9%)	OR = 2.09 [0.71, 6.16] (*p* = 0.282)
	Microvascular disfunction	28 (100%)	23 (100%)	51 (100%)	

**Table 3 jcm-10-02284-t003:** Relationship between LBBB-QRS complex duration and magnitude of coronary impairment in patients with ischemic heart disease by macrovascular involvement.

Subset	*N*	Mean ± *SD*	Med (min:max)
QRS (ms) (Shapiro—Wilk normality test: *p* < 0.001)			
(total)	70 (100.0%)	134.714 ± 8.96	130.00 (120.00:160.0)
Coronary lessions (Wilcoxon rank sum test with continuity correction: *p* = 0.675)			
yes	19 (27.1%)	133.684 ± 6.84	130.00 (120.00:150.0)
no	51 (72.9%)	135.098 ± 9.67	130.00 (120.00:160.0)
Coronary lessions4 (Kruskal—Wallis rank sum test: *p* = 0.415)			
none	51 (72.9%)	135.098 ± 9.67	130.00 (120.00:160.0)
univascular	9 (12.9%)	132.222 ± 6.67	130.00 (130.00:150.0)
bivascular	5 (7.1%)	132.000 ± 8.37	130.00 (120.00:140.0)
trivascular	5 (7.1%)	138.000 ±4.47	140.00 (130.00:140.0)

**Table 4 jcm-10-02284-t004:** The correlation between the presence of LBBB, its duration, and the structural changes of LV evaluated by echocardiography.

Subset	*N*	Mean ± *SD*	Med (min:max)
QRS (ms) (Shapiro—Wilk normality test: *p* < 0.001)			
(total)	70 (100.0%)	134.714 ± 8.96	130.00 (120.00:160.0)
IVS dissynchronism (Wilcoxon rank sum test with continuity correction: *p* = 0.414)			
Yes	68 (97.1%)	134.853 ± 9.06	130.00 (120.00:160.0)
No	2 (2.9%)	130.000 ± 0.00	130.00 (130.00:130.0)
IVS size (mm) (Spearman’s rank correlation rho: R = −0.091, *p* = 0.452)			
(total)	70 (100.0%)	11.671 ± 1.64	12.00 (8.00:16.0)
Diastolic LV diameter (mm) (Spearman’s rank correlation rho: R = 0.476, *p* < 0.001)			
(total)	70 (100.0%)	56.486 ± 9.31	55.50 (38.00:87.0)
Systolic LV diameter (mm) (Spearman’s rank correlation rho: R = 0.499, *p* < 0.001)			
(total)	70 (100.0%)	41.529 ± 11.53	40.00 (20.00:77.0)

**Table 5 jcm-10-02284-t005:** The presence of negative prognostic factors (age, sex, and heart failure) in patients with LBBB.

Variable	Details	F	M	Total	Statistics
Gender		35 (50%)	35 (35%)	70	
Heart failure		23 (65.7%)	27(77.1%)	50 (71.4%)	OR = 0.57 [0.20, 1.63] (*p* = 0.428)
NYHA classification	II III	9 (39.1%) 14 (60.9%)	14 (51.9%) 13 (48.1%)	23 (46.0%) 27 (54.0%)	OR = 0.60 [0.19, 1.84] (*p* = 0.407)
OR: odds-ratio [95% CI] and *p* value from Fisher test					
**Subset**		***N***	**Mean ± *SD***	**Med (min:max)**	
Age (Shapiro-Wilk normality test: *p* = 0.617)		70 (100.0%)	67.014 ± 8.89	67.00 (47.00:88.0)	
HF (Welch Two Sample *t*-test: *p* = 0.370)					
Da		50 (71.4%)	67.600 ± 9.13	68.50 (47.00:87.0)	
Nu		20 (28.6%)	65.550 ± 8.30	65.50 (50.00:88.0)	
NYHA classification (Welch Two Sample *t*-test: *p* = 0.633)					
II		23 (46.0%)	66.913 ± 9.96	68.00 (47.00:84.0)	
III		27 (54.0%)	68.185 ± 8.51	69.00 (52.00:87.0)	

HF: heart failure; NYHA: New York Heart Association.

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
