# Peer review of "Major Left Bundle Branch Block and Coronary Heart Disease—Are There Any Differences between the Sexes?"

_jcm, 2021, doi:10.3390/jcm10112284_

Round 1

Reviewer 1 Report

  1. Were all the echocardiography pictures reviewed and measured by the investigator? I am a echo and MR reader. Just want to make sure that the measurements were counter checked by the author. The author should mention how the measurements were made for LVEDD , LVESV and for EF?
  2. 2. Again when you are comparing LBBB and heart failure, you have to define heart failure? Is it systolic heart failure alone or was HfpEF also included?.
  3. 3. Also you have to mention how microvascular disease is diagnosed in your study? Is it based on CFR measurements by PET scan or diagnosis of exclusion when cath is negative for CAD?
  4. 4. Also how did you select the 70 patients with LBBB? Was this by chart review and random sampling and retrospective analysis? This has to be mentioned.
  5. 5. Minor typo errors.--line 76, line 122

Author Response

Thank you very much for your review, which I find very useful in significantly improving my article. I have provided point-by-point answers to your observations below.

  1. Were all the echocardiography pictures reviewed and measured by the investigator? I am a echo and MR reader. Just want to make sure that the measurements were counter checked by the author. The author should mention how the measurements were made for LVEDD , LVESV and for EF?

Response 1: All the echocardiography pictures are stored on our local ultrasound machine. I have performed all the measurements included in the article myself. The LVEDD, LVESD were measured from parasternal long-axis view and the EF was calculated using Simpson’s biplane formula.

  1. Again when you are comparing LBBB and heart failure, you have to define heart failure? Is it systolic heart failure alone or was HfpEF also included?

Response 2: The diagnosis of Heart Failure was established taking into acount the symptoms, clinical exam, echocardiography parameters (EF, diastolic dysfunction) and NT-proBNP values. Patients with HfpEF were also included.

  1. Also you have to mention how microvascular disease is diagnosed in your study? Is it based on CFR measurements by PET scan or diagnosis of exclusion when cath is negative for CAD?

Response 3: The diagnosis of microvascular coronary artery disease was based on a negative coronary angiography in a patient with angina. Coronary flow reserve was not measured.

  1. Also how did you select the 70 patients with LBBB? Was this by chart review and random sampling and retrospective analysis? This has to be mentioned.

Response 4: Patients with LBBB were identified when routine ECG in recorded at the moment of admission into our department and included in the study after signing the informed consent.

  1. Minor typo errors.--line 76, line 122

Response 5: Thank you for pointing them. I will correct the errors.

Reviewer 2 Report

All of my track changes and comments deleted when I went to make my comments anonymous, so I apologize for those not being present. Fortunately, all of my grammatical changes and suggestions were saved. Please look over the entire document to see the changes I made.

Author Response

Thank you very much for your review, I really appreciate your hard work. I find it very useful in significantly improving my article. I fully agree with the changes that you recommended.

Best regards,

Diana Gurzau.